# Effects of Heat Treatment on the Structural and Functional Properties of *Phaseolus vulgaris* L. Protein

**DOI:** 10.3390/foods12152869

**Published:** 2023-07-28

**Authors:** Chaoyang Li, Yachao Tian, Caihua Liu, Zhongyou Dou, Jingjing Diao

**Affiliations:** 1National Coarse Cereal Engineering Technology Research Center, Heilongjiang Bayi Agricultural University, Daqing 163319, China; lzy885400@163.com (C.L.); dou265258@163.com (Z.D.); 2School of Food Science and Engineering, Qilu University of Technology, Jinan 250353, China; tianyachaopaper@163.com; 3College of Food Science, Northeast Agricultural University, Harbin 150030, China; 17686963949@163.com

**Keywords:** *Phaseolus vulgaris* L. protein, heat treatment, protein structural, protein function, digestibility

## Abstract

The paper presents the effect of heat treatment at 80 °C at different times (3, 5, 7, and 9 min) on the structural and functional properties of *Phaseolus vulgaris* L. protein (PVP, bean protein powder). Surface and structure properties of PVP after heat treatment were analyzed using a Fourier transform infrared spectrometer (FTIR), a fluorescence spectrophotometer, a visible light spectrophotometer, a laser particle size analyzer, and other equipment. The secondary structure and surface hydrophobicity (H_0_) of PVP changed significantly after heat treatment: the β-sheet content decreased from 25.32 ± 0.09% to 24.66 ± 0.09%, the random coil content increased from 23.91 ± 0.11% to 25.68 ± 0.08%, and the H_0_ rose by 28.96–64.99%. In addition, the functional properties of PVP after heat treatment were analyzed. After heat treatment, the emulsifying activity index (EAI) of PVP increased from 78.52 ± 2.01 m^2^/g to 98.21 ± 1.33 m^2^/g, the foaming ability (FA) improved from 87.31 ± 2.56% to 95.82 ± 2.96%, and the foam stability (FS) rose from 53.23 ± 1.72% to 58.71 ± 2.18%. Finally, the degree of hydrolysis (DH) of PVP after gastrointestinal simulated digestion in vitro was detected by the Ortho-Phthal (OPA) method. Heat treatment enhanced the DH of PVP from 62.34 ± 0.31% to 73.64 ± 0.53%. It was confirmed that heat treatment changed the structural properties of PVP and improved its foamability, emulsification, and digestibility. It provides ideas for improving PVP’s potential and producing new foods with rich nutrition, multiple functions, and easy absorption.

## 1. Introduction

Providing a wealth of proteins, minerals, and fiber, kidney beans (*Phaseolus vulgaris* L.) are recommended by the FAO/WHO [1]. PVP is a vital plant protein source that contains a variety of essential amino acids and provides the opportunity to develop nutritional products [1]. Unlike animal proteins, PVP is lactose-free, cholesterol-free, and low in fat, which makes it a healthy source of protein [2]. Research has shown that PVP contains anti-nutritional factors such as tannin, which reduce its digestibility and functionality [3]. Anti-nutritional factors greatly limit plant protein digestibility and usage, so improving its functional properties and industrial utilization is an important problem that needs to be addressed as soon as possible.

Protein structural and functional can be modified through protein modification to meet production, manufacturing, and commercial needs [4]. Protein modification can be achieved by chemical [5], physical [6], or enzymatic [7] methods. This allows for an alteration in protein physical properties, such as solubility, stability, and activity [6]. Chemical modification is a common and effective method of changing protein structure and properties. However, chemically-modified protein has shortcomings such as insufficient safety verification, residual chemical substances, and low production efficiency [8]. Although enzymatic modification avoids the shortcomings of chemical modification, enzymatic modification has the disadvantages of strict biological enzyme reaction conditions, easy inactivation, difficult recovery, and high cost [8]. As the most common physical modification method—thermal modification—it will not cause safety problems and can reduce production costs significantly [9]. Thus, thermal modification is favored by researchers and food manufacturers.

There is evidence that moderate heat treatment will unfold the protein structure and expose the internal amino acid residues, thus impacting the structural and functional properties of the protein [10]. Ye et al. [11] revealed that heat or UHT-treated milk is easier for humans to digest and absorb than fresh milk, because heat-treated milk forms more fragmented and brittle structures. According to Yang et al. [12], when the heating temperature is elevated, the tertiary structure of the protein depolymerizes, improving the surface hydrophobicity (H_0_) and emulsification properties. Because of its low cost and good results, heat treatment has become a common method of modification. At present, there are few studies on the effect of heat treatment time on the structure and properties of PVP. In order to improve the digestibility and functional properties of PVP, it is important to study the effect of heat treatment times.

Thus, this experiment studies the influence of heat treatment at different times (3, 5, 7 and 9 min) on PVP structural and functional properties. PVP structural properties were characterized by particle size, turbidity, secondary structure, H_0_, disulfide bond, and thermal stability. The functional properties of PVP were characterized by water and oil holding properties, solubility, emulsifying properties, foaming properties, and in vitro digestibility. The regulation mechanism of heat treatment time on PVP was analyzed, and the heat treatment conditions that were beneficial to improving the functional properties of PVP were determined. This was to establish a theoretical basis for PVP research and application to different food systems.

## 2. Materials and Methods

### 2.1. Materials

PVP (bean protein powder, isoelectric point 4.2) was accessed from Heilongjiang Agriculture Co., Ltd. (Heilongjiang, China). Corn oil was purchased from Macklin Biochemical Technology Co., Ltd. (Shanghai, China). Nile blue, Nile red, isopropanol, 8-aniline-1-naphthalenesulfonate (ANS), and other chemicals were accessed from Beijing Dingguo Changsheng Biotechnology Co., Ltd. (Beijing, China). All other reagents were analytical grade.

### 2.2. Heat Treatment

Based on the research method used by Ellouze et al. [10], 5 g of PVP was dissolved in 100 mL of phosphate buffer solution (5 mM, pH 7.0). An appropriate amount of sodium azide (0.02%, *w*/*v*) was added to the protein buffer, stirred for 2 h, and stored at 4 °C overnight. Then, 50 mL of completely dissolved PVP solution was put in a 100 mL sealed bottle. Following this, it was placed in a water bath at 80 °C for 3, 5, 7, and 9 min, respectively. After heat treatment, all samples were cooled to room temperature in an ice bath and preserved at 4 °C until use. After the remaining protein solution was stored at −20 °C for at least 24 h, it was freeze-dried for 24 h using an LGJ-10 freeze dryer from Songyuan Huaxing Biotechnology Co., Ltd. (Beijing, China). The freeze-drying area of LGJ-10 freeze dryer was 0.12 m^2^, the water catchment capacity was 3–4 kg/24 h, the ultimate vacuum degree was ≤5 par, the material in the tray was 1.2 L (the material thickness is 10 mm), and the cold trap temperature was ≤−56 °C. The detection of samples was carried out at room temperature.

### 2.3. Particle Size of PVP

Based on the methods by Guo et al. [1], which was appropriately modified, distilled water was used to dissolve the 1 g PVP sample. The sample needed to be clear and transparent. Then, the particle size distribution was measured using the S3500-laser particle size analyzer. To avoid sample swelling, measurements were performed immediately after sample preparation.

### 2.4. Turbidity

To investigate the effect of heat treatment time on PVP turbidity, spectroscopic techniques were used. The absorbance of all samples was detected by a V-5000 visible light spectrophotometer (Yuanxi Instrument Co., Ltd., Shanghai, China) with a wavelength detection range of 325–1000 nm and a spectral bandwidth of 4 nm. The PVP samples with different heat treatment times were prepared into 2 mg/mL solutions respectively. The methods by Yucel et al. [13] was used with minor modifications. First, the PVP sample was poured into a glass cuvette with an optical path length of 1 cm. The optical density (OD) of the sample at 400 nm was recorded at 25 °C as a measure of the system’s turbidity. All samples were measured three times.

### 2.5. Secondary Structure

The method by Tang [14] and Rafe [15] was modified: the secondary structure of protein was detected by Bruker Vertex 70 Fourier Transform Infrared Spectrometer (FTIR), signal-to-noise ratio: 50,000:1, sampling rate: 80 spectra/second, measurement spectrum area: 30,000–10 cm^−1^, step scan-time resolution: 5 ns. The scanning wavenumber range of PVP samples was 4000–400 cm^−1^.

### 2.6. Fluorescence Intensity

Referring to the method of Jiang et al. [16], the F-7000 Fluorescence Spectrophotometer (RMS signal-to-noise ratio of 800, scanning speed of 60,000 nm/min) was used to detect the fluorescence emission spectrum of PVP samples. Then, the PVP samples were diluted to 0.2 mg/mL in phosphate buffered saline. Acquired spectra at excitation wavelengths were 295 nm and 300–400 nm.

### 2.7. Surface Hydrophobicity (H_0_)

Using 1-anilino-8-naphthalenesulfonic acid (ANS) as a hydrophobic fluorescent probe, the H_0_ of PVP samples was measured by referring to the method of Musa et al. [17]. A mixture of PVP (4 mL, 0.2–1.0 mg/mL) and ANS (20 mL, 8 mM) was kept at room temperature and without light, and a fluorometer was used to measure PVP fluorescence intensity at 280 and 350 nm. H_0_ was estimated based on a linear regression equation of initial slope versus fluorescence intensity and protein concentration (mg/mL).

### 2.8. Free Sulfhydryl (FS) and Disulfide Bonds (DB)

The approach used by Tang et al. [14] was minimally adjusted. The DTNB method (5,5′-dithiobis-(2-nitrobenzoic acid)) was utilized to measure the FS quantity of PVP samples. First, PVP samples were prepared in dH_2_O to 0.01 g/mL, then 50 μL of the sample mix was diluted with 200 μL buffer (Tris-Gly Urea). The DTNB was added and reacted for 20 min. Below is the FS calculation formula:(1)$$\begin{array}{c} \mathrm{FS}\left(\frac{\mu mol}{g}\right)=\frac{73.53\;\times \;D{\;\times \;A}_{412}}{C} \end{array}$$
where: 73.53—DTNB extinction coefficient; *D*—dilution factor; *A*_412_—sample absorbance; *C*—PVP concentration, mg/mL.

Then, 1.0 mL of PVP sample (0.01 g mL^−1^) was added to 15.0 mL of Tris-Gly urea buffer, mixed with 0.3 mL of β-mercaptoethanol, and reacted at 25 °C for 1 h. During the reaction, 100 mL of 12% trichloroacetic acid (TCA) was added, followed by centrifugation for 10 min at 4000 rpm. Then, the precipitate was dissolved in 20 mL Tris-Gly Urea buffer, mixed with 0.2 mL DTNB (4 mg/mL), and reacted at 25 °C for 20 min. The formulas for calculating the total sulfhydryl (TS) content and DB content are as follows:(2)$$TS\left(\mu mol/g\right)=\frac{73.53\times A_{412}}{C}$$
(3)$$DB\left(\mu mol/g\right)=\frac{TS-FS}{2}$$
where: 73.53—DTNB extinction coefficient; *A*_412_—PVP absorbance; *C*—PVP dose, mg/mL.

### 2.9. Water Holding Capacity (WHC) and Oil Absorption Capacity (OAC)

For determining the WHC and OHC of PVP samples, the experimental approach by Liu et al. [18] was modified. PVP (0.5 g) and 5 mL of distilled water or oil were added to a pre-weighed centrifuge tube and incubated at 25 °C for 30 min. Then, they were centrifuged at 2000 rpm for 30 min to remove free water or oil, and excess water or oil was drained from the sample with filter paper. As a result, WHC and OAC were obtained by calculating the following:(4)$$WHC(g/g)=\frac{M2-M1}{m}$$
(5)$$OAC(g/g)=\frac{M2-M1}{m}$$
where: *m*—the weight of PVP sample, *M*2—total weight of PVP sample; centrifuge tube and water or oil; *M*1—the weight of sediment and centrifuge tube after centrifugation.

### 2.10. Solubility

The solubility determination of PVP samples with different heat treatment times refers to the research methods of Wang et al. [19]. First, a 5 mg/mL PVP solution was prepared, centrifuged at 5000 rpm for 10 min, and the supernatant was the sample solution to be tested. The formula for calculating the solubility of PVP is as follows:(6)$$Solubility=\frac{m1}{m2}\times 100\%$$
where: *m*1—PVP weight in supernatant; *m*2—total PVP weight.

### 2.11. Emulsifying Properties of PVP

#### 2.11.1. Emulsion Preparation

To prepare a coarse emulsion, 15 mL of the 5% (*m*/*v*) PVP solution was used as the water phase, followed by 5 mL of corn oil, homogenized at 10,000 rpm for 2 min. The emulsion was obtained after the coarse emulsion was further homogenized by a high-pressure homogenizer.

#### 2.11.2. Particle Size and ζ-Potential

The emulsion particle size and ζ-potential were determined according to Tian et al. [20]. The particle size and ζ-potential of the samples were measured by a Malvern NANO ZS90 laser particle size analyzer (measurement range: particle size and molecular size: 0.3 nm–5.0 μm, ζ-potential: 3.8 nm–100 μm, molecular weight: 9800 Da–20 MDa). Each sample was measured three times.

#### 2.11.3. Confocal Laser Scanning Microscopy (CLSM)

The microstructure of the emulsions was observed using an LSM 900 model confocal laser scanning microscope (Carl Zeiss AG, Oberkochen, Germany). Referring to the method of Rafe et al. [21] with a slight modification, CLSM was used to observe PVP emulsion droplet distribution. Dissolve Nile blue and Nile red in isopropanol to prepare staining solutions with concentrations of 1% (*m*/*v*) and 0.1% (*m*/*v*) respectively. Then, 200 μL of the emulsion was diluted to 1 mL, 55 μL of Nile blue and 50 μL of Nile red was added, mixed well, and placed in the dark for 0.5 h, using CLSM to observe and take pictures.

#### 2.11.4. Emulsifying Activity Index (EAI) and Emulsifying Steadiness Index (ESI)

Based on Pearce et al. [22], EAI and ESI of PVP were measured. Corn oil (3.75 mL) and protein solution (3% *w*/*v*, 21.25 mL) were mixed and homogenized at 10,000 rpm for 1 min. After homogenization, 200 μL of the emulsion was taken at 0 and 10 min and added to 1.8 mL of 0.1% sodium dodecyl sulfate (SDS), and its absorbance was measured at 500 nm. Calculated as follows:(7)$$EAI\;(m^{2}/g)=\frac{2\times 2.303\times N\times A_{0}}{C\times \varphi \times L\times 10,000}$$
(8)$$ESI/min=\frac{10\times A_{0}}{A_{0}-A_{10}}$$

In the formula: *N* stands for the ratio of dilution; *φ* represents the volume ratio of the oil phase; *L* represents the cuvette thickness, cm; *C* represents the protein mass concentration, mg/mL; *A*_0_ and *A*_10_ represent the absorbance at 0 min and 10 min, respectively.

### 2.12. Foaming Ability (FA) and Foam Stability (FS)

The FA and FS of PVP can be measured using the method by Wang et al. [23] with a few adjustments. First, 20 mL of PVP sample solution (3% *w*/*v*) was homogenized at 13,500 rpm for 1 min, and the mixture was immediately transferred to a graduated cylinder. FA and FS are calculated as follows:(9)$$FA\left(\%\right)=\frac{V_{0}}{20}\times 100$$
(10)$$FS\left(\%\right)=\frac{V_{30}}{V_{0}}\times 100$$

In the formula: *V*_0_—volume of foam at 0 min; 20—PVP sample solution initial volume, mL; *V*_30_—volume of foam at 30 min.

### 2.13. Degree of Hydrolysis (DH)

According to the method by Ge et al. [24], we produced simulated gastric fluid (SGF) and simulated intestinal fluid (SIF). First, 0.5 g of PVP samples treated with different heat treatment times were mixed with 10 mL of SGF, and the pH was adjusted to 3.0 with 3.0 M HCl. The mixture was reacted at 37 °C and 150 rpm for 2 h. After the simulated gastric digestion, 10 mL of SIF was added to the mixture and reacted at 37 °C and 150 rpm for 2 h. After the reaction, the mixture was centrifuged (10,000× *g*, 30 min, 4 °C), and the supernatant was collected and filtered with a 0.45 μm filter membrane.

The DH of PVP after simulated gastrointestinal digestion was measured by the OPA method with reference to the method of Ge et al. [24]. The specific method is as follows: First, 80 mg o-phthalaldehyde (OPA), 3.81 g disodium tetraborate decahydrate, 100 mg sodium dodecyl sulfate (SDS), and 88 mg dithiothreitol (DTT) were dissolved in 100 mL of deionized water to obtain the OPA solution. Then, 20 mg of L-serine was weighed and dissolved in 200 mL of deionized water, and recorded as the serine standard solution. The digested supernatants were diluted 10 times, and the protein concentration was measured by BCA kit, which was recorded as C_Sample_. Next, 3 mL of OPA solution was mixed with 400 μL of diluted digestion solution, serine standard solution, and deionized water, and reacted accurately for 2 min, and the absorbance value was recorded at 340 nm. The absorbance value of the sample was recorded as OD_Sample_, the absorbance value of serine was recorded as OD_Standard_, and the absorbance value of deionized water was recorded as OD_Blank_. The DH of PVP can be calculated by the following formula:(11)$$Serine\;{NH}_{2}=\frac{{OD}_{Sample}-{OD}_{Blank}}{{OD}_{Standard}-{OD}_{Blank}}\times 0.9516\times \frac{1}{C_{Sample}}$$
(12)$$h=\frac{Serine{NH}_{2}-\beta }{\alpha }$$
(13)$$DH\left(\%\right)=\frac{h}{h_{tot}}$$

In the formula, *α* = 0.970 mequv/g, *β* = 0.342 mequv/g, *h_tot_* = 7.8 mequv/g.

### 2.14. Statistical Analysis

An average value was obtained by multiplying three measurements by standard deviation (SD). Data were analyzed with IBM SPSS software (version 20.0, IBM, Chicago, IL, USA) using analysis of variance (ANOVA) with *p* < 0.05 as statistically significant.

## 3. Results and Discussion

### 3.1. Particle Size of PVP Analysis

PDI reflects the particle size distribution range. Higher PDI values indicate a wider range of particle sizes and therefore an increased potential for heterogeneity. In Figure 1, the PDI values of PVP show a tendency to increase and then decrease with prolonged heat treatment. The main peak position of PVP particle size distribution without heat treatment was 100–1000 nm. As heat treatment time increases, the position of the main peak of the particle size distribution of PVP moves to the left to about 100 nm. The single peak becomes a double peak. After 7 min of heat treatment, PVP formed a peak with a small particle size and the largest PDI. This is because the heat treatment destroys the PVP intermolecular bonds [25]. The PVP partly disintegrates, but the particle size distribution range increases and particle size decreases. When the heat treatment time is 9 min, PVP particle size increases. When proteins are overheated, they aggregate, resulting in larger particles of PVP [1]. The above results show that different heat treatment times will have different effects on PVP particle size and distribution range.

### 3.2. Turbidity Analysis

The turbidity of the solution refers to the degree of hindrance to light induced by the suspended solids in the solution. Figure 2 shows that turbidity was represented by the optical density value of the PVP sample at 400 nm. Researchers have found that turbidity in solutions was not only affected by suspended matter and cavity matter but also by particle size, shape and surface reflection and scattering [26]. The turbidity value of PVP without heat treatment was 0.208 ± 0.012, and the turbidity of PVP after heat treatment was significantly improved. Heat treatment time from 3 min to 7 min, the turbidity value of PVP increased from 0.378 ± 0.015 to 0.824 ± 0.022. This is because as the heat treatment time increases, the denaturation of PVP molecules intensifies, the particle size of PVP molecules decreases, and the distribution is uneven, and the molecules are easily cross-linked and aggregated through covalent bonds or non-covalent bonds to form thermal aggregates of small molecules, leading to an increase in turbidity [27]. When the heat treatment time was 9 min, PVP turbidity was 0.643 ± 0.017. This is because when heat treatment time is too long, PVP molecules aggregate to form larger particles, resulting in reduced turbidity [28]. The above results show that heat treatment breaks the intermolecular bonds of PVP and makes the particle size distribution uneven, thereby changing the turbidity. Heat treatment affects PVP properties, which means the particle size and shape can be controlled by heat treatment. Moreover, heat treatment can also be used to modify the material’s surface properties [29,30].

### 3.3. Secondary Structure Analysis

PVP’s secondary structure was determined by FTIR. Related studies have shown that proteins’ secondary structure composition was a significant factor affecting their functional properties [31]. Table 1 shows that, compared with untreated PVP, heat treatment significantly changed its secondary structure composition. With the prolonged heat treatment time, the content of β-sheet, α-helix, and β-turn of PVP decreased first and then increased, while the content of random coil increased first and then decreased. It appears that proper heat treatment can exacerbate PVP structure disorder. Heat treatment time was 7 min, the β-sheet content of PVP was the least, while the random coil content was the most, indicating that the PVP structure was the loosest and disordered at this time. During the heating process, hydrogen bonds that stabilize the secondary structure are destroyed, resulting in partial denaturation and unfolding of protein molecules, which destroys the folded protein structure [32]. Heat treatment time was 9 min, the content of β-sheets improved, the content of random coils decreased, and the protein structure began to order again. Heat treatment that lasts for too long causes the protein structure to unfold, which increases hydrophobic interactions, and partially denatured protein molecules to form protein aggregates via intermolecular interactions, resulting in an increase in protein order as a result [1,33]. According to these results, heat treatment changed the secondary structure of PVP by partially denaturing and opening it.

### 3.4. Fluorescence Analysis

Studies have shown that the maximum absorption wavelength (λ_max_) of the fluorescence spectrum is related to the environment of aromatic amino acid residues inside the protein, and the tertiary structure of the protein can be reflected by observing the fluorescence spectrum [34]. Figure 3 shows that PVP’s maximum fluorescence intensity without heat treatment is 323.41 A. U, and the λ_max_ is 330.43 nm. As the heat treatment time was extended, the maximum fluorescence intensity of PVP first decreased, then increased. In this study, the wavelength shifted from 330.43 nm to 335.08 nm, indicating that aromatic amino acid residues in the protein changed from hydrophilic to hydrophobic environments [16,35]. After heat treatment for 7 min, the maximum fluorescence intensity of PVP was the lowest, 201.34 A. U. This is because the PVP structure is opened by heat treatment, and the exposure of the internal aromatic groups with luminescent function leads to fluorescence quenching, thereby reducing the fluorescence intensity [36,37].

### 3.5. H_0_ Analysis

The surface hydrophobicity of proteins is related to aromatic amino acid residues and aliphatic amino acid residues [38]. H_0_ indicates the ability of hydrophobic groups to contact the surface of an aqueous environment [39]. H_0_ is closely related to the functional properties of proteins, such as emulsifying [40], gelling [41], foaming properties [42], etc. Figure 4 shows that compared with PVP without heat treatment, heat treatment significantly increases the H_0_ value of PVP. The H_0_ value of PVP without heat treatment is 65.87 ± 1.65. The H_0_ of PVP after heat treatment for 3 min is 84.94 ± 1.52. Heat treatment time was extended from 0 min to 3 min, H_0_ increased by 28.95%. Heat treatment time was 7 min, the highest H_0_ value of PVP was 108.68 ± 2.63. Compared with heat treatment for 0 min, the H_0_ of PVP increased by 64.99% after heat treatment for 7 min. Due to the increased heat treatment time, the folded structure of PVP opens more, exposing more hydrophobic groups within, increasing contact between the protein and surface water, resulting in a higher H_0_ value [43]. Heat treatment time was 9 min, the H_0_ value of PVP decreased to 102.21 ± 1.85. Compared with heat treatment for 0 min, the H_0_ of PVP increased by 55.19% after heat treatment for 9 min. The reason for this is that unfolded or denatured proteins develop an unstable structure when heated too long, forming aggregates of varying degrees through intermolecular cross-linking and hydrophobic interactions [44,45]. Guo et al. [1] found a similar pattern with polymerized kidney bean protein that was thermally soluble.

### 3.6. FS and DB Analysis

FS and DB in proteins can maintain spatial conformation and endow proteins with certain functional properties. Interconversion between FS and DB is the basis of protein functional property [46]. Table 2 shows that compared with PVP without heat treatment, heat treatment significantly enhances the FS content of PVP. It also significantly reduces DB. The FS content of PVP increased initially and then decreased with increased heat treatment time. In addition, the DB content first decreased and then increased. When the heat treatment time was 7 min, the FS content was the highest at 0.772 ± 0.007 mol/g, the DB content was the lowest at 0.396 ± 0.017 μmol/g. The DB in PVP is destroyed by heat treatment, resulting in a decrease in DB content [47]. During this process, the PVP structure loosens and becomes disordered, exposing the FS inside, increasing its content [48]. When the heat treatment time was 9 min, the FS content decreased to 0.601 ± 0.018 μmol/g, and the DB content increased to 0.553 ± 0.105 μmol/g. This is due to the heat treatment time being too long, which leads to FS being exchanged for DB, which results in the decrease in FS and an increase in DB [49]. It appears that different heat treatment times result in different degrees of unfolding of PVP and exposure of internal sulfhydryl groups, thereby altering the content of DB and FS.

### 3.7. WHC and OAC Analysis

The WHC measures how much water a protein can absorb and hold onto, while the OAC measures how much oil a protein can absorb and hold onto [50]. These two measurements are important in determining the functional properties of a protein, such as its solubility, binding ability, and stability [51]. Figure 5 shows that the WHC and OAC of PVP without heat treatment were 3.58 ± 0.11 g/g and 2.39 ± 0.08 g/g, respectively. Different heat treatments increased the WHC and OAC of PVP. In response to the lengthening of the heat treatment time, the WHC and OAC of PVP increased initially, then declined. When the heat treatment time was 7 min, the highest WHC and OAC of PVP were 4.19 ± 0.10 g/g and 3.25 ± 0.07 g/g, respectively. This indicates that heat treatment is an effective method for enhancing the water holding capacity and oil absorption capacity of PVP [52]. There is a possibility that heat treatment breaks non-covalent bonds between PVP molecules, such as hydrogen bonds and hydrophobic bonds [53,54]. As a result, the PVP structure stretches, and the PVP-water interaction force increases. At the same time, the random curl content in PVP molecules increases. In addition, water molecules intercalate more easily into protein voids, and both the WHC and OAC of PVP increase. When the heat treatment time was 9 min, the WHC and OAC of PVP decreased to 3.96 ± 0.10 g/g and 2.91 ± 0.05 g/g. At this time, cross-linking occurs between protein molecules, resulting in a decrease in water and oil holding capacity [55]. Consequently, the heat treatment of PVP improves its WHC and OAC properties, making it an ideal choice for many applications.

### 3.8. Solubility Analysis

Hydrophobic and hydrophilic groups on the surface of proteins determine their solubility, which is closely related to their foaming and emulsifying properties [56]. The hydrophobic groups, which are nonpolar, are attracted to oils and fats, while the hydrophilic groups, which are polar, are attracted to water. This creates an interface between the oil and water molecules, which can then be used to form foam or emulsions [57]. According to Figure 6, PVP’s solubility without heat treatment is 66.52 ± 0.69% and increases significantly after heat treatment. When the heat treatment time increased from 3 min to 7 min, PVP solubility increased from 74.56 ± 0.42% to 82.15 ± 0.49%. When heat treatment time increased to 9 min, PVP solubility decreased to 79.24 ± 0.59%. A moderate heating treatment may enhance the ability of PVP to combine with water because it opens the structure and exposes the hydrophilic groups [58]. If the heat treatment time is too long, the thermal aggregation between PVP molecules increases the intermolecular force, reducing PVP’s ability to combine with water, thus reducing its solubility [59]. Bogahawaththa et al. [60] reported that specific heating temperature and time combinations can have distinct effects on protein solubility. This is similar to the way that cooking a steak requires knowing the right combination of time and temperature to obtain the desired result: a perfectly cooked steak. Too little time or too low a temperature will leave you with an undercooked steak, while too much time or too high a temperature will lead to an overcooked steak.

### 3.9. Emulsification Analysis of PVP

#### 3.9.1. Particle Size and ζ-Potential Analysis of PVP Emulsion

The ability of proteins to emulsify is one of their most important functions. Pickering emulsions prepared using proteins as emulsifiers have been widely used in the fields of food, medicine, and cosmetics [20]. Figure 7 illustrates the effect of heat treatment time on the particle size (A) and ζ-potential (B) of PVP Pickering emulsion. Figure 7A shows that the heat treatment results in peaks in the particle size distribution of PVP emulsions, which indicates that the particle size distribution is uneven after the heat treatment. The peak of the particle size distribution of the PVP emulsion gradually shifted left when the heat treatment time was extended from 3 min to 7 min. The smallest particle size of PVP emulsion is achieved after 7 min of heat treatment. The main peak of the particle size distribution of PVP emulsion shifts to the right again when the heat treatment time exceeds 7 min, and the particle size of the emulsion increases. This may be due to the increase in PVP’s H_0_ as well as its enhanced emulsifying ability following moderate heat treatment, thereby reducing its particle size. After heat treatment for 9 min, PVP is easy to cross-link to form aggregates, and the particle size of emulsion droplets increases and the distribution is uneven [1]. Therefore, it is important to find the optimal heat treatment time for PVP in order to maximize its emulsifying ability.

The ζ-potential is used to evaluate the stability of emulsion systems. It is determined by the surface charge state of the emulsifier particles in the emulsion system. The greater the absolute value of the potential, the more stable the emulsion system [61]. Based on Figure 7B, heat treatment significantly increases the absolute potential of the PVP emulsion, and its change trend is consistent with its particle size distribution. When the heat treatment time is 7 min, the absolute value of the emulsion potential is the largest. It is possible that a certain degree of heat treatment breaks the PVP molecular bonds, exposes the wrapped charged groups, and increases their surface charge [62]. The surface charge of PVP will decrease when the heat treatment time is too long, because intermolecular aggregation is easy to occur and part of the charges will be re-wrapped [63]. According to the above results, heat treatment changed both the surface charge state and the exposure degree of charged groups inside the PVP molecule, resulting in a change in its absolute value.

#### 3.9.2. CLSM Analysis

CLSM is an advanced technology used to directly observe the morphology structure and size of PVP emulsion droplets as well as the distribution of lipid particles in the droplets [64]. Figure 8 shows that the distribution of emulsion droplets prepared by heat-treated PVP is more uniform and dispersed. It can be seen in Figure 8(4) that PVP subjected to 7 min heat treatment produces the smallest and most uniform emulsion droplets. CLSM results of emulsions prepared by PVP with different heat treatment times are consistent with emulsion particle size distribution results. As a result of the CLSM study, we were able to confirm our previous speculations about how heat treatment time affects PVP.

#### 3.9.3. EAI and ESI Analysis

The EAI of protein refers to the stable oil–water interface area per unit mass of protein, and the ESI of protein refers to the ability of protein to maintain emulsion stability within a predetermined time [65]. The EAI of a protein mainly depends on the diffusion and adsorption of the protein at the oil-water interface [38]. Figure 9 shows the effect of heat treatment time on the EAI and ESI of PVP. According to Figure 9, heat treatment significantly improves the EAI and ESI of PVP. As the heat treatment time increased, the EAI and ESI of PVP increased and then decreased. The EAI and ESI of PVP without heat treatment are 78.52 ± 2.01 m^2^/g and 174.35 ± 3.45 min, respectively. PVP has the highest EAI (98.21 ± 1.33 m^2^/g) and ESI (221.72 ± 3.91 min) when the heat treatment time is 7 min. Studies have shown that increasing proteins’ hydrophobicity can improve their ability to bind to water–oil interfaces [66]. Therefore, moderate heat treatment leading to the exposure of PVP hydrophobic groups is the main reason for the significant increase in EAI and ESI. EAI and ESI of PVP decrease when the heat treatment time exceeds 7 min. The reason for this is that when the heat treatment time is too long, electrostatic repulsion offsets hydrophobicity, resulting in the formation of PVP flocs, which lowers EAI and ESI values [1].

### 3.10. FA and FS Analysis

The protein dissolves in the aqueous phase to form a cohesive layer of protein around the gas/air droplet for foam capability [38]. In foamed foods such as cakes, biscuits, toffee, and ice cream, the foam can provide a pleasant appearance and a delicate taste. FA and FS are significant parameters for evaluating whipped foods’ stability [67]. The FA and FS of PVP are shown in Figure 10. As shown in Figure 10, heat treatment improves both FA and FS of PVP. The FA and FS of PVP without heat treatment were 87.31 ± 2.56% and 53.23 ± 1.72%, respectively. When the heat treatment time was 3 min, the FA and FS of PVP were 90.17 ± 3.19% and 56.34 ± 1.92%, respectively. PVP has the highest FA (95.82 ± 2.96%) and FS (58.71 ± 2.18%) when the heat treatment time is 7 min. FA increased by 6.27% when heating time increased from 3 min to 7 min. FS increased by 4.21% when heating time increased from 3 min to 7 min. Overheating PVP will cause excessive denaturation, which will reduce the contact between PVP and the air-water interface, resulting in decreased foam stabilization ability. It agrees with the results Lajnaf et al.’s [68] study on how heat affected purified α-lactalbumin from camel milk. In the above results, proper heat treatment improved the FA and FS of PVP, which made it more suitable for foaming foods. Heat treatment also provides better solubility and thermal stability for PVP, making it an excellent stabilizer for food products. Furthermore, PVP offers a wide range of functional properties, making it a valuable ingredient in the food industry.

### 3.11. DH Analysis

Figure 11 shows the final DH in simulated gastrointestinal digestion in vitro. The DH of PVP was increased by heat treatment at different times. The DH value of PVP without heat treatment was 62.34 ± 0.31%, and the highest DH value of PVP was 73.64 ± 0.53% when the heat treatment time was 7 min. When the heat treatment time exceeds 7 min, the DH value drops slightly again. DH increased by 18.13% when the heat treatment time increased from 0 min to 7 min. Studies have found that β-sheets are negatively correlated with protein digestibility. Appropriate heat treatment (in this paper refers to heat treatment for 7 min) reduces β-sheets in the secondary structure of PVP, which increases the exposure to protease cleavage sites and increases the DH value of proteins. When PVP was heat-treated for 9 min, the exposed protease cleavage sites decreased and the DH value decreased. The above results show that moderate heat treatment can significantly increase the DH value of PVP.

## 4. Conclusions

In summary, heat treatment of PVP at 80 °C for 3, 5, 7, and 9 min all improved the structural and functional properties of PVP. After heat treatment for 7 min, the particle size of PVP was the smallest and the content of sulfhydryl groups and H_0_ were the highest. When the heat treatment time increased from 3 min to 7 min, the solubility, WHC, OAC, FA, FS, and DH of PVP increased by 10.18%, 10.55%, 29.48%, 6.27%, 4.21%, and 13.73%, respectively. When the heat treatment time exceeds 7 min, the structural and functional properties of PVP weaken, but they are better than they would be without heat treatment. In this experiment, treatment at 80 °C for 7 min was the most effective time–temperature combination to improve PVP structure and functional properties. We have confirmed through experiments that heat treatment changes the structure properties of PVP and improves its foamability, emulsification, and digestibility. An idea for modifying PVP efficiently and at a low cost is presented by this result. Moreover, this study offers an approach to applying PVP to traditional food, foam foods, and emulsion foods.

## Figures and Tables

**Figure 1 foods-12-02869-f001:**
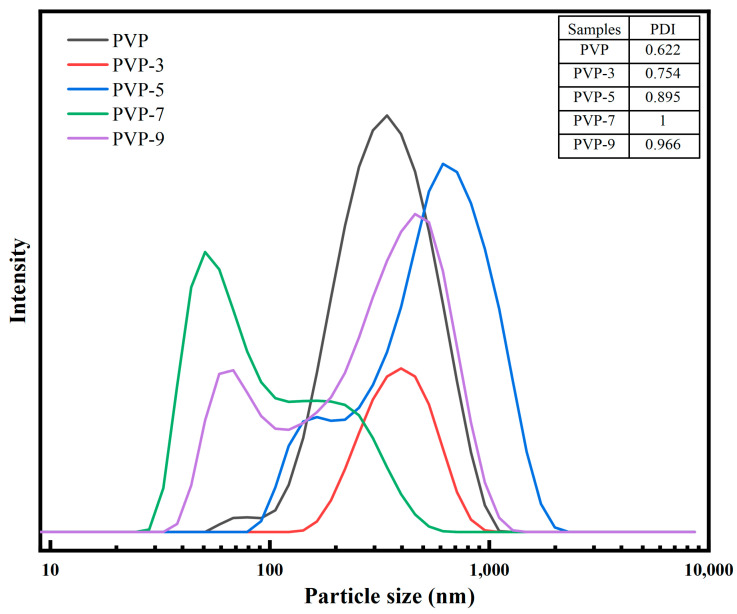
Effect of heat treatment time on particle size of PVP.

**Figure 2 foods-12-02869-f002:**
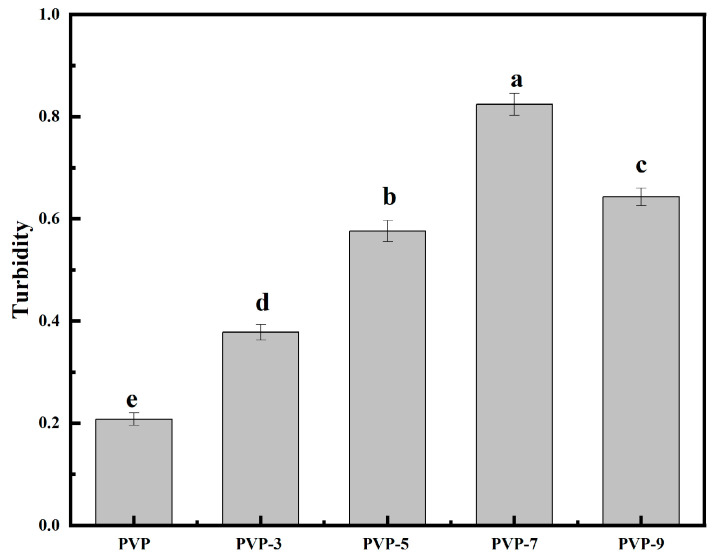
Turbidity of PVP influenced by heat treatment time. Significant differences (*p* < 0.05) are indicated by different letters.

**Figure 3 foods-12-02869-f003:**
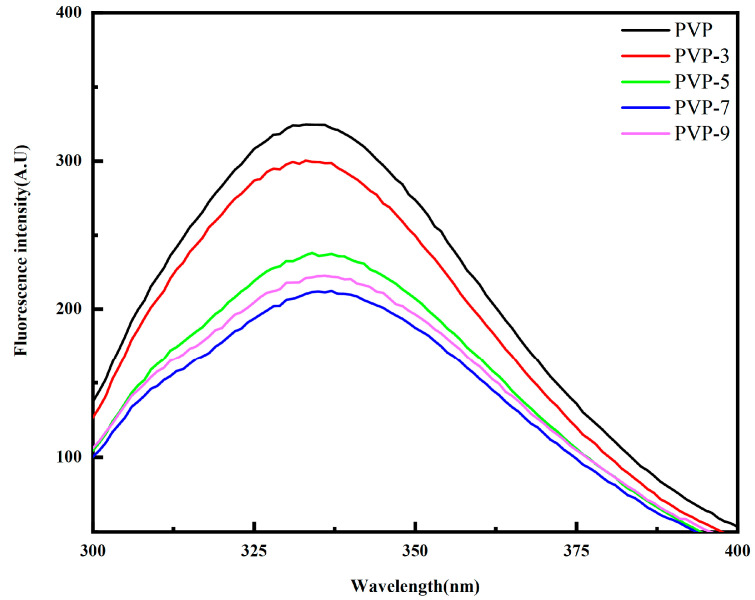
Effect of heat treatment time on fluorescence intensity of PVP.

**Figure 4 foods-12-02869-f004:**
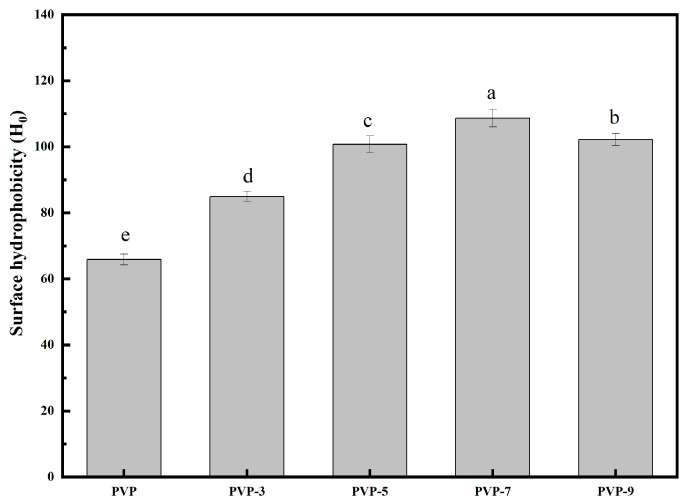
Effect of heat treatment time on H_0_ of PVP. Significant differences (*p* < 0.05) are indicated by different letters.

**Figure 5 foods-12-02869-f005:**
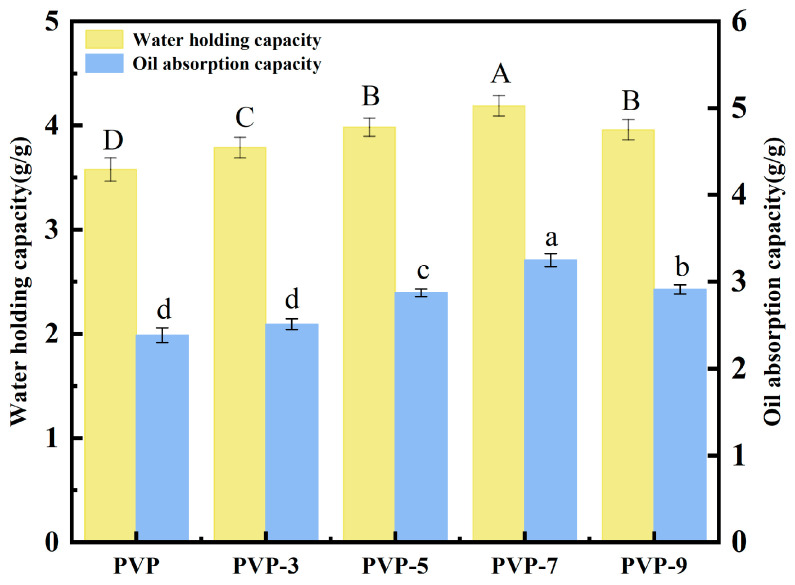
Effect of heat treatment time on WHC and OAC of PVP. Significant differences (*p* < 0.05) are indicated by different letters.

**Figure 6 foods-12-02869-f006:**
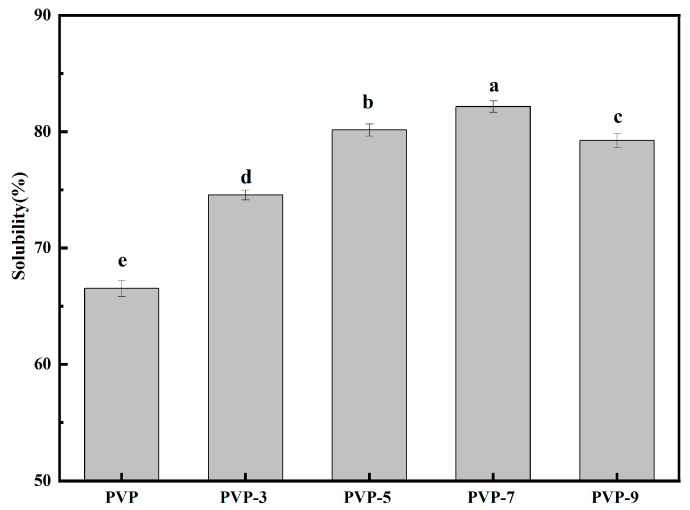
Effect of heat treatment time on the solubility of PVP. Significant differences (*p* < 0.05) are indicated by different letters.

**Figure 7 foods-12-02869-f007:**
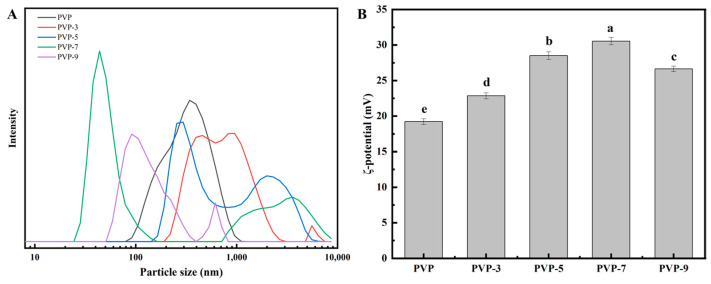
Effect of heat treatment time on the particle size (**A**) and ζ-potential (**B**) of PVP Pickering emulsion. Significant differences (*p* < 0.05) are indicated by different letters.

**Figure 8 foods-12-02869-f008:**
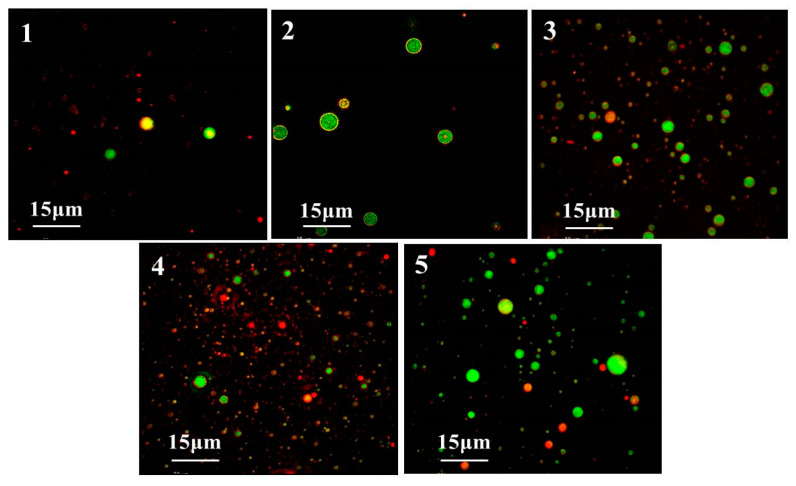
Effect of heat treatment time on the CLSM of PVP emulsion ((**1**): PVP, (**2**): PVP-3, (**3**): PVP-5, (**4**): PVP-7, (**5**): PVP-9).

**Figure 9 foods-12-02869-f009:**
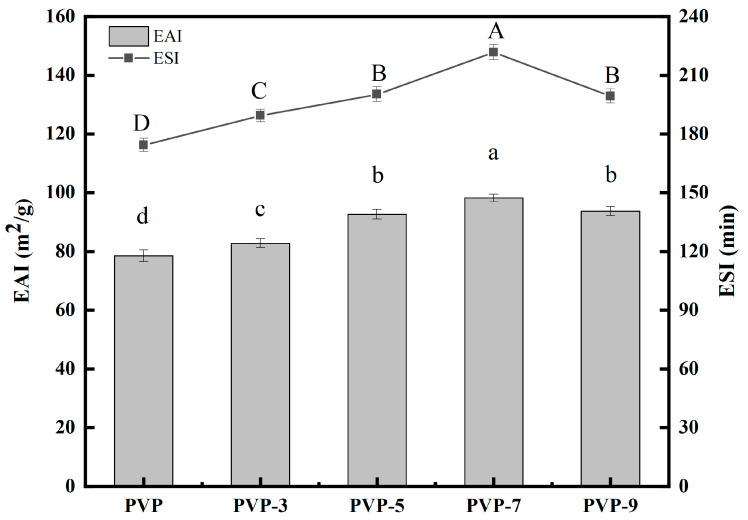
Effect of heat treatment time on the EAI and ESI of PVP. Significant differences (*p* < 0.05) are indicated by different letters.

**Figure 10 foods-12-02869-f010:**
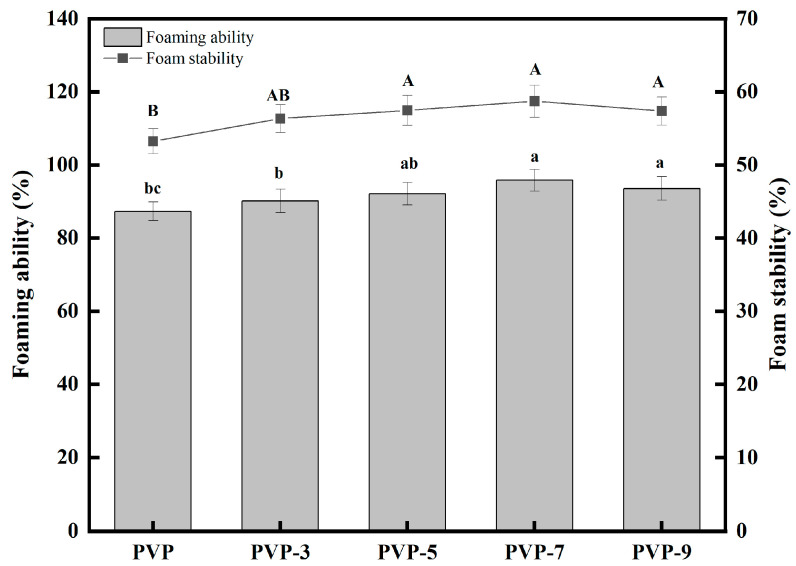
Effect of heat treatment time on the FA and FS of PVP. Significant differences (*p* < 0.05) are indicated by different letters.

**Figure 11 foods-12-02869-f011:**
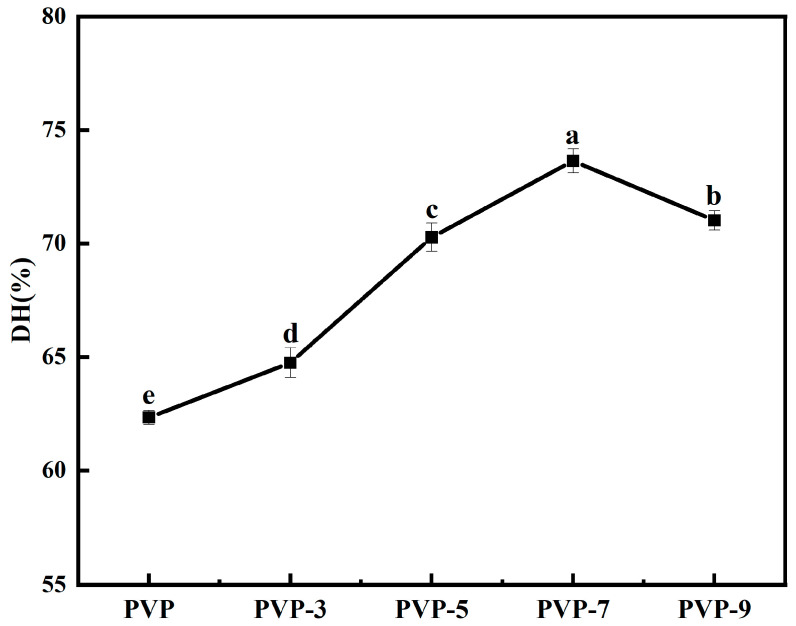
Effect of heat treatment time on the DH of PVP. Significant differences (*p* < 0.05) are indicated by different letters.

**Table 1 foods-12-02869-t001:** Effect of heat treatment time on the secondary structures of PVP.

Sample	β-Sheet (%)	α-Helix (%)	β-Turn (%)	Random Coil (%)
PVP	25.32 ± 0.09 ^a^	21.27 ± 0.12 ^a^	15.92 ± 0.14 ^a^	23.91 ± 0.11 ^d^
PVP-3	25.02 ± 0.08 ^bc^	20.86 ± 0.09 ^b^	15.21 ± 0.09 ^b^	24.38 ± 0.13 ^c^
PVP-5	24.79 ± 0.14 ^c^	19.39 ± 0.08 ^c^	12.94 ± 0.12 ^c^	25.12 ± 0.15 ^bc^
PVP-7	24.66 ± 0.09 ^cd^	18.98 ± 0.11 ^d^	12.05 ± 0.08 ^e^	25.68 ± 0.08 ^a^
PVP-9	25.09 ± 0.11 ^b^	19.37 ± 0.07 ^c^	12.54 ± 0.07 ^d^	25.29 ± 0.11 ^b^

Note: Values in each column with different superscript letters are significantly different (*p* < 0.05).

**Table 2 foods-12-02869-t002:** Effect of heat treatment time on the free sulfhydryl and disulfide bonds of PVP.

Sample	Free Sulfhydryl (μmol/g)	Total Sulfhydryl (μmol/g)	Disulfide Bonds (μmol/g)
PVP	0.586 ± 0.009 ^c^	2.107 ± 0.565 ^a^	0.760 ± 0.284 ^a^
PVP-3	0.588 ± 0.005 ^c^	1.699 ± 0.175 ^b^	0.555 ± 0.088 ^b^
PVP-5	0.603 ± 0.004 ^b^	1.476 ± 0.078 ^d^	0.437 ± 0.040 ^d^
PVP-7	0.772 ± 0.007 ^a^	1.564 ± 0.032 ^c^	0.396 ± 0.017 ^e^
PVP-9	0.601 ± 0.018 ^b^	1.708 ± 0.208 ^b^	0.553 ± 0.105 ^b^

Note: Values in each column with different superscript letters are significantly different (*p* < 0.05).

## Data Availability

Data is contained within the article.

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
