# Peer review of "Effects of Heat Treatment on the Structural and Functional Properties of Phaseolus vulgaris L. Protein"

_foods, 2023, doi:10.3390/foods12152869_

Round 1

Reviewer 1 Report

The manuscript examined the structural and functional properties of PVP as affected by different temperatures and times. Although it seems a proper work, it lacks form some important information. Firstly, the pH of protein is important and should be clearly mention in the paper at which temperature these properties were measured. Secondly the type of PVP is more important and its specification for future works is required. Some other comments are given as follows:

Line 19-21: Please do not use abbreviation in the abstract like as EAI, FS, FA, OPA.

Line 62: please correct the English structure.

Line 81: please provide more details concerning the PVP.

Line 84: protein dispersion. Is the protein completely dissolved?

Line 87: please provide the specification of freeze drying.

Line 99: please provide the specification of spectrophotometer.

Line 104: It should be stated more details about the FTIR. Use this information from the work “Effect of thermal treatment on chemical structure of β-lactoglobulin and basil seed gum mixture at different states by ATR-FTIR spectroscopy” such as resolution and etc.

Line 107: specification of fluorescence spectrophotometer?

Line 191: rpm instead of r/min.

Line 167: the specification of particle size analyzer is required.

Line 169-174: use the work “Rheological and structural properties of β-lactoglobulin and basil seed gum mixture: Effect of heating rate” for the CLSM work and specify the conditions. Furthermore, Confocal Laser scanning Microscopy is correct, so correct the Abbreviation in this section.

Line 185: use superscript for A0 and A10.

Line 199, 201: rpm instead of r/min.

Results and discussion: it is strongly recommended to use the work “Functional properties of rice bran protein isolate at different pH levels, 2016” in this section to have comparison with other plant proteins such as rice bran protein.

The comments are given.

Reviewer 2 Report

This manuscript investigates kidney bean proteins and their structural and functional properties in response to heat treatment. Overall, the manuscript organization, discussion, and presentation of the results are very good. However, I have several points that will improve the content and help clarify some points if addressed by the authors. Most importantly, heat treatment is a conventional/traditional technique to improve the techno-functional properties of proteins. Therefore, the use of “innovative” does not apply to the manuscript. If the authors are willing to use this term, it is necessary to explain the innovative aspect of this study.

Abstract:

    The authors use the plant name in the title and the abstract. It is unclear if the beans are referred to or what form they were obtained.  
    Too many abbreviations in the abstract without identifying what they are. Abbreviations should be given in parentheses when they are first used, and then abbreviations should be used.  
    The use of innovative is not applicable, therefore, it needs to be removed.  
    Please clarify if the beans are utilized or the plant. Were they dried or wet?  

Intro:  

5.     Line 63-72: How did the authors select these times? Based on what? Any preliminary analysis to designate the heat treatment times, please explain.

Materials and Methods:  

6.     Please give more information about the protein, such as powder form, isoelectric point, etc.  

7.     The authors explain their methodology by giving directions rather than full sentences such as “take 5 g” instead of 5 g of protein was dissolved in a buffer solution. The authors can prepare a flow chart showing their steps in the methods they use, which would be more appropriate with this language.  

Results:  

8.     Line 225: As this part includes discussion, name this section as “Results and Discussion”  

9.     Line 232-233: instead of “heat treatment time prolongation”, use “prolonged heat treatment”  

10.  Line 274: “Prolongation of” should be replaced by “prolonged”  

11.  Table 1: For PVP, the statistical results seem incorrect if the analysis is for each column, as indicated in the note. If we look into the results in the columns separately, the random coil should have been “a” instead of “d”. There is no “c,” and if 15.92 is “a,” so does 23.91, given that 21.27 and 25.32 are “a” as well. Otherwise, the statistical analysis was made for the entire table, not individual columns. Please check this and implement the correction.  

12.  Line 311-323: It may be helpful to give a percentage change in H0 and other parameters when the heat treatment time is increased from 3 to 5 to 7 to 9 min.  

13.  Line 329: Please replace “functional activity,” as this terminology does not sound right. Maybe functional property or performance? It is better to keep the terminology consistent with the literature.

14.  In Figure 9, the unit for the EAI is m2 per g, which is too large for an interfacial area. The results seem off. Please explain.  

15.  Line 485, what do you refer to as “appropriate heat treatment”? Could you specify?  

Conclusion:  

16.  Line 503: Instead of “process parameter”, the use of “time-temperature combination” is more appropriate.  

17.  More importantly, the percent changes in the techno-functional properties for the increased time increments would be helpful. For example, foaming ability increased xx% when heating time increased from 3 min to 7 min. The readers would benefit from such useful information.  

18.  Lastly, in lines 506-508, the use of “innovative” is not applicable, as explained earlier- heat treatment is a traditional approach to such applications.  

Still needs a proofreading. 

Round 2

Reviewer 1 Report

Comments are replied correctly

Author Response

Thank you so much for your positive review.

Reviewer 2 Report

The authors have addressed all the points raised by the referee. 

Minor editing would be good to eliminate the repetition of similar words in the same sentence or the following sentences. 

Author Response

Comments: Minor editing would be good to eliminate the repetition of similar words in the same sentence or the following sentences.  Answer: I appreciate the valuable comments you provided that make our manuscript more perfect. We have revised our manuscript based on your suggestions. Please see lines 43-45,107-109,271-273,279,378-379,423-425 and 506-507.